An architecture for non-linear discovery of aggregated multimedia document web search results

http://orcid.org/0000-0001-6014-1356 Khan Abdur Rehman arkhan@cs.qau.edu.pk
http://orcid.org/0000-0002-3453-7979 Rashid Umer umerrashid@qau.edu.pk
Saleem Khalid
Ahmed Adeel
Department of Computer Sciences, Quaid-i-Azam University , Islamabad , Pakistan
Gauch Susan
Electronic publication date: 2021 Apr 21
Publication date: 2021
Volume: 7
Electronic Location ID: e449
Received 2020 Dec 1; Accepted 2021 Mar 1
Copyright: © 2021 Khan et al.
Copyright year: 2021
Copyright holder: Khan et al.
License: This is an open access article distributed under the terms of the Creative Commons Attribution License, which permits unrestricted use, distribution, reproduction and adaptation in any medium and for any purpose provided that it is properly attributed. For attribution, the original author(s), title, publication source (PeerJ Computer Science) and either DOI or URL of the article must be cited.
License URL: https://creativecommons.org/licenses/by/4.0/

Keywords: Multimedia, Web, Search engine, Exploratory search, Information discovery, Relational aggregated search

Funding: The authors received no funding for this work.

==============================
The recent proliferation of multimedia information on the web enhances user information need from simple textual lookup to multi-modal exploration activities. The current search engines act as major gateways to access the immense amount of multimedia data. However, access to the multimedia content is provided by aggregating disjoint multimedia search verticals. The aggregation of the multimedia search results cannot consider relationships in them and are partially blended. Additionally, the search results’ presentation is via linear lists, which cannot support the users’ non-linear navigation patterns to explore the multimedia search results. Contrarily, users’ are demanding more services from search engines. It includes adequate access to navigate, explore, and discover multimedia information. Our discovery approach allow users to explore and discover multimedia information by semantically aggregating disjoint verticals using sentence embeddings and transforming snippets into conceptually similar multimedia document groups. The proposed aggregation approach retains the relationship in the retrieved multimedia search results. A non-linear graph is instantiated to augment the users’ non-linear information navigation and exploration patterns, which leads to discovering new and interesting search results at various aggregated granularity levels. Our method’s empirical evaluation results achieve 99% accuracy in the aggregation of disjoint search results at different aggregated search granularity levels. Our approach provides a standard baseline for the exploration of multimedia aggregation search results.

Introduction

Traditionally, the web contains only the textual content (Bianchi-Berthouze et al., 2003). The progressive easy access to the internet has transformed the web into an infinitely complex virtual organism consisting of immense multimedia content (Batrinca & Treleaven, 2015). The format of the information is now extremely varied. The individual bits of data coming from blogs, articles, web services, picture galleries, etc., are resulting in exponential growth of multimedia data on the web (Batrinca & Treleaven, 2015; Rashid & Bhatti, 2017). The web is becoming the most ubiquitous platform ever since its birth and has increased in both quantity and quality (Taheri et al., 2018). In 2009, less than 1 petabyte of digital data was created daily (Kumar & Ogunmola, 2020). It grew to approximately 2.5 exabytes in 2012 and reached 4.4 zettabytes in 2013. On the web, the digital data in different formats created, replicated, and consumed exponentially (Oussous et al., 2018). It is doubling every 2 years. By 2015, digital data grew to 8 zettabytes, and the volume of data will reach 40 zettabytes by the end of 2020 (Oussous et al., 2018).

Keywords-based general web search engines have made early efforts to provide access to multimedia information (Lewandowski, 2008). These search engines required a user to enter one or a few keywords, and the search engines produced the relevant results in a short time (Lewandowski, 2008). Kerne & Smith (2004) first discussed a new search paradigm called information discovery. They elaborated discovery as a long journey of search that begins with a vague description of a problem, may have an articulated set of criteria during which a searcher specify a query and evaluate the returned information surrogates, and may continue iteratively by re-evaluating the result sets and forming a sense of desired results. Marchionini (2006) gives the same idea in a broader perspective by categorizing the search paradigm into an exploratory by incorporating not only lookup searches but learning and investigation activities. Adequate support in the users’ search leads to the discovery of new information items.

In contrast, many current search systems assume an exploratory search process as a series of homogeneous steps of submitting a query and consulting search results. Research in information seeking has shown that users go through discrete phases in their search journey, from exploring and identifying preliminary information to refining and narrowing their information needs and search strategies to finalize the search. It is reported as a highly complex problem bridging the different areas of information seeking, interactive information retrieval, and user interface design (Gäde et al., 2015). Moreover, the increasing amount of heterogeneous content on the web has transformed user needs from simple lookup-based queries to broader exploratory queries, requiring the diverse heterogeneous contents to satisfy the desired information needs (Rashid & Bhatti, 2017).

Several studies indicate that more intricate tasks resulted in a diversity of the information sought and more varied approaches to information seeking (Campbell, 2013). Today, to find interesting multimedia content, an enormous number of users use search engines (Deldjoo et al., 2018). It has changed users’ information need from textual to multi-modal (audio, image, and video) searching. Approximately 40–50% of users engage in dynamic and unplanned nature of web multimedia searches (Tseng, Tjondronegoro & Spink, 2009). When the information need is ambiguous and dynamic (e.g., in exploratory search), people often consult more multimedia search results (Bron et al., 2013). The need for multimedia documents, in this case, increases to 58% (Tseng, Tjondronegoro & Spink, 2009).

As human information needs and search tasks become complex, the users have to collect and assemble information from diverse information sources. The goal is to compose the most appropriate responses to the tasks at hand in the form of multimedia documents (Kopliku, Pinel-Sauvagnat & Boughanem, 2014). A multimedia document is a collection of co-existing heterogeneous multimedia objects sharing the same semantics (Rashid & Bhatti, 2017). Users prefer aggregation of useful multimedia information residing in diverse sources through unified interfaces (Kopliku, Pinel-Sauvagnat & Boughanem, 2014; Rashid & Bhatti, 2017). Similarly, the user interface presenting aggregated contents encouraged participants to view more diversified sources from the search results, and 75% of the participants found this blended approach more comfortable to use (Sushmita, Joho & Lalmas, 2009). The user click-through rate analysis reported approximately 33% on augmented multimedia artifacts and nearly 55% multimedia artifacts were found relevant and useful during information exploration activities (Sushmita et al., 2010). Overall, users explore the multimedia contents 78% of the time to answer their dynamic information needs (Kopliku et al., 2011). Based on recent user behavior in complex information needs, we can easily forecast even more increasing multimedia artifacts consumption from the users in satisfaction of complex information needs and discovering information.

Aggregating disjoint verticals provide access to diverse multimedia documents. A vertical is defined as a specialized assembly of same-typed documents (Bakrola & Gandhi, 2016). This assembly can be media-specific or domain-specific. The former may include media types (e.g., video, blog, image, etc.). The latter may consist of verticals (e.g., travel, shopping, news, etc.). The aggregation process consists of either cross-vertical Aggregated Search (cvAS) or Relational Aggregated Search (RAS). The (cvAS) ignores the relation during retrieval and aggregation of multimedia content. The (RAS) considers the relationships in the multimedia information. Despite the key-role of aggregation in bridging the modality gap, this area of research only received limited attention in the past (Achsas & Nfaoui, 2019). Without substantial creativity, this area of research will soon be abandoned. Our discovery approach aims to bring innovation and creativity in this area of search. We envision bridging the modality gap and shortcomings of current search engines, allowing users to discover multimedia information by aggregating disjoint verticals.

The contributions of our solution are threefold. Firstly, we presented a creative search results aggregation technique using state-of-the-art semantic analysis. Secondly, we enhanced the current search engine shortcomings in information exploration and discovery activities by augmenting non-linear information seeking patterns. Thirdly, we bridged the information modality gap by encoding the search results in various representations. Our proposed solution is the first to address all of the stated challenges of information aggregation, exploration, and discovery.

The rest of the discussion is organized as follows. We discuss the related work in “Related Work”. We highlight the deficiencies in the existing approaches and motivation behind this research in “Problem & Motivation”. We provide the theoretical foundation and formalization of our proposed approach in “Architecture Design: Definition, Formalization & Instantiation”. We present the implementation of the architecture in “Architectural Implementation”. We discuss the experimental results in “Results”. Finally, we compare our approach with state-of-the-art and conclude our discussion in “Comparison & Discussion” and “Conclusion & Future Work”, respectively.

Related work

Theoretical background and frameworks

According to Kerne et al., (2008), information discovery tasks require finding and collecting relevant information elements; filtering the collected elements; developing an understanding of the found elements and their relationships. The overall goal is assembling information and connecting answers to open-ended questions (Kerne et al., 2008). It is a multidisciplinary approach and is built on anomalous states of knowledge (Belkin, Oddy & Brooks, 1982), berry-picking (Bates, 1989), psychological relevance (Harter, 1992), exploratory search (White et al., 2006), information foraging (Pirolli & Card, 1999), information seeking (Marchionini, 1997) and sensemaking (Baldonado & Winograd, 1996; Russell et al., 1993).

During a task performed by a user, the lack of information triggers the requisite for the information needs. The recognized information needs refer to as an anomalous state of knowledge (Belkin, Oddy & Brooks, 1982). During the recognized anomalous state of knowledge, the users refer to the information retrieval systems to initiate the information seeking journey (Marchionini, 1997). During this journey, the user picks relevant information analogous to an organism picking berries in the forest scattered on the bushes; they do not come in bunches. One must select them one at a time (Bates, 1989). Similar to this analogy, the user has to forage for the information and pick items from information patches giving information scent the most (Pirolli & Card, 1999). Information scents are the cues that help the user in making sense of the provided information. It can be augmented by sensemaking activity. It involves making sense of the data during data analysis, searching for representation, and encoding the data to answer specific task-oriented questions (Russell et al., 1993). This whole journey can incorporate lookup search, learning, and investigation activities, resulting in a non-linear search pattern.

To do this non-linear search of the information successfully, researchers must leverage their skills and experience to develop search systems that actively engage searchers using semantics, inherent structure, and meaningful categorization (White & Roth, 2009). In general, the user cannot precisely specify what is needed to resolve recognized information anomaly (Belkin, Oddy & Brooks, 1982). It often results in the shortcomings of the existing retrieval systems in a scenario where the user cannot correctly formulate information need expression resulted in low precision of the retrieval systems (Belkin, Oddy & Brooks, 1982). In such case, the users’ information needs are not fully satisfied by a single final retrieved set, but by a series of selections of individual bits of information at each stage of the ever-modifying search strategies (Bates, 1989). Hence, these tasks require more recall over precision (Tablan et al., 2015). We must consider the user perspective of information relevance, taking into account how effective the topic of the information retrieved matches the subject of interest and how to represent a piece of information that induces a change in the users’ cognitive state (Harter, 1992).

Our proposed solution encodes the multimedia search results semantically by aggregating them in multimedia documents. These documents allow users to pick the most suitable collection of information sufficing their information need the most. Furthermore, we provide multimedia document groups, analogous to patches of information, allowing the user to forage for the information patches giving the most information scent. We increased the information scent for multimedia documents and groups by summarizing the data inside them. Semantically aggregated disjoint multimedia verticals provide the conceptualization of multimedia documents and groups. Furthermore, we augment the non-linear information searching and seeking pattern by instantiating a non-linear graph comprising various granularity levels of search results and proximally similar multimedia content links.

From federation to aggregation & diversification

Traditional research mostly centered on assisting users in providing relevant multimedia information federated from various sources and information providers. Koh et al. (2006) provided discovery of search results by dispatching the user query to multiple search engines and extracting the relevant pieces of text snippets and images snapshot on the user interface. Similarly, Sushmita, Lalmas & Tombros (2008) provided a digest-based information exploration approach by collecting various pieces of multimedia information from a variety of sources and encapsulating them in the form of a digest. Afterward, researchers identified the modality gap of information with enormously increasing heterogeneous content on the web, which hindered the information exploration. Hence, the first idea of search results aggregation was presented in a workshop at ACM SIGIR 08 conference (Kopliku, 2009). Later on, Sushmita, Joho & Lalmas (2009) advanced this idea towards the blending and evaluation of disjoint multimedia verticals into the web search results (Sushmita et al., 2010).

Information aggregation is now widely recognized, considered a bridge that narrows the information modality gap and fosters information exploration. Meanwhile, search engines are also starting to adopt a similar approach in their presentation of the search results (Bakrola & Gandhi, 2016). The progress in multimedia retrieval presented another challenge in deciding the optimal choice and position of vertical in the search engine result page and was explicitly labeled as a vertical prediction problem. Bakrola & Gandhi (2016) provided a solution to this challenge by using implicit feedback of the user in the form of several clicks and then using a support vector machine classifier to predict the most suitable vertical sufficing the given user information needs.

Nowadays, the most common and popular commercial web search engines such as Baidu (https://www.baidu.com/), Bing (https://www.bing.com/), Google (https://www.google.com/), Yahoo! (https://www.yahoo.com/), Yandex (https://yandex.com/) etc., are blending some vertical-specific results, assembled from the other data sources into the linear ranked list of standard results. Moreover, a recent trend focuses on the information diversification aspects of the information (Taramigkou, Apostolou & Mentzas, 2017). It usually involves integrating more diverse verticals (e.g., other than image, news, video, and web). This diversification may include the integration of verticals from social media, shopping, movies & dramas, maps, songs, etc. However, this integration of the verticals is mostly partial-blended (Rashid & Bhatti, 2017). The relationship between the multimedia artifacts inside each disjoint vertical is often ignored.

The aggregation approaches can be implmented as a multimedia or multimodal system. The multimedia system uses single modality (usually textual metadata associated with multimedia content) to bridge the multiple modalities (Benavent et al., 2013). The multimodal systems incorporates multiple modalities to provide access to multimedia content (Benavent et al., 2013). The aggregation of the multimedia artifacts demands a better solution to enhance user interaction with the search results. It is essentially a very broad problem and answered by (RAS) techniques. The researchers leveraged some effort in (Rashid & Bhatti, 2017), they performed (RAS) using textual, visual, and acoustic descriptors of the multimedia contents. However, this multimodal aggregation was provided using a generic similarity measure for each modality and ignored the semantics relationships in aggregated multimedia documents. In (Achsas & Nfaoui, 2018), researchers presented a stacked auto-encoders model for multimedia aggregation of the disjoint verticals. However, their research addresses a small aspect of the aggregation and ignores information exploration perspectives.

Renovation in information exploration data-models & semantic web

The current practices for information exploration include presenting the aggregated verticals as a linear list (Tablan et al., 2015). It is due to a lack of data-model flexibility. Initially, using the semantic web techniques and ontologies was perceived as a promising start. For instance, Tablan et al. (2015) presented an open-source semantic framework providing indexes and searches using document structure, metadata, annotations, and semantics through linked open data. The architecture supported both; information seeking and exploration & discovery tasks by two distinct user interfaces designed, respectively.

Similarly, Lisena et al. (2017) developed a modern web application for music exploration and discovery using semantic RDF graphs to establish links between entities and relationships among them. Khalili et al. (2017) used inference techniques on the semantic linked open data to produce notably unique information fostering discovery. However, due to scalability challenges in exploiting the whole web of Linked Data limits the practicality of this aspect (Elzein et al., 2018).

Similarly, in (Kanjanakuha, Janecek & Techawut, 2019), researchers provided semantic data representation in a hyperbolic tree format. Their framework consists of a 3-layers hyperbolic tree-based modal approach that takes the input in the form of keywords from the user. The information is then presented in the form of a graph. The 3-layer approach divides the complexity of information in each layer. It reduces the confusion caused by information overload and enhances significant interaction and navigation. Similar to our proposed approach, their graph data-model provides highlighting, node describing, zooming, panning, and linking functionalities.

More researchers are presently making an effort to provide a generalized approach to exploring and discovering multimedia artifacts on the web. It includes mixing different aspects of data-model, diverse information aggregation, and visualization. For instance, in (Rashid & Bhatti, 2017), researchers provided a generalized framework for relational aggregation of the multimedia artifacts belonging to disjoint sets using a graph-based visualization and exploration of a multimedia search result space. Similarly, in (Zhang et al., 2018), researchers developed a discovery engine for artificial intelligence research. Their architecture crawls the web, downloads the research papers from various journal websites, and performs full-text indexing using a cosine similarity measure. It builds a similarity-based network having similarity links in documents. Users’ stars, clicks, and tweets are primarily used to reinforce the graph’s essential connections.

However, the past approaches focused on using a domain-specific dataset and data-model using generic textual and visual similarity metrics. We establish a data-model using the semantics that exists inside the data. Specifically, we semantically found part-of or containment relationships in the multimedia artifacts. Moreover, we also instantiate similarity links among the multimedia artifacts that allow navigation to similar multimedia artifacts. We opt to keep the data-model as generic as possible without relying on domain knowledge and explicit feedback, making our solution implementable on a wide range of domains.

Problem & motivation

The users’ complex information-seeking behavior is modeled as a non-linear journey requiring adequate support during the navigation of the information space (Ruotsalo et al., 2018). Users’ forage for the information (Bates, 1989). Their complex information needs are not sufficed through the current ideology of returning the most precise information in response to the given queries (Russell-Rose & Tate, 2012). Instead, users pick the most interesting items like barries from various patches of information, providing more information scent ratio to the effort required for examining the information. It results in a non-linear information searching pattern of users (Russell-Rose & Tate, 2012).

The search engines are more tuned towards simple lookup searches favoring precision over recall (Tablan et al., 2015). However, even though they have recognized the users’ multimedia information needs and started to blend some vertical-specific results assembled from the other data sources (Bakrola & Gandhi, 2016). The current practices of presenting information in a linear ranked list of standard results limit information exploration (Rashid & Bhatti, 2017). Furthermore, the integration of the verticals is mostly partial-blended (Rashid & Bhatti, 2017), which may suffice in simple lookup searches when a user knows what to look for; however, this strategy inadequately support complex information exploration and discovery tasks (Tablan et al., 2015). These tasks go beyond simple keyword-based queries. Users often have difficulties in information need expression, and they usually are dynamic (Ruotsalo et al., 2015a). Such tasks require more recall over precision and diversity of information sources (Tablan et al., 2015). It challenges the current practices of displaying the search results belonging to different verticals as disjoint sets (Rashid & Bhatti, 2017).

On the other hand, the search engines remain almost the same as they were about a decade ago. There exist numerous problems (P) with current search engines. The Fig. 1 shows the difference between the Google Search Engine Results Page (SERP) back in 2010 (Sullivan, 2020) and now in 2020 (Google, 2020). The verticals are integrated as disjoint components (P1). The relationships between multimedia objects are ignored (P2). The information presented is still displayed as linear lists (P3). This presentation of the general search engines’ information may suffice for simple lookup tasks but lacks adequacy for complex exploratory and discovery tasks (Klouche et al., 2015). These tasks require increased recall over precision (P4), information scent (P5), and sensemaking (P6). The existing exploration approaches’ deficiencies demand a better mechanism to encode and present the multimedia information for discovery (P7).

Figure 1 Comparison of Google SERP between 2010 (left) and 2020 (right).

(A) Enhanced Snippet, (B) Question Answer Vertical, (C) Videos Vertical, (D) Web Vertical, (E) Images and Related Searchers. Screenshot credit: Google and the Google logo are trademarks of Google LLC.

Architecture design: definition, formalization & instantiation

Existing techniques are usually specific to a problem and employed on a particular dataset. Many researchers consider one side of the discovery, such as information diversification, visualization, data-modal etc., and ignore the other factors highlighted in the previous section. To the best of our knowledge, a generalized multimedia search results discovery mechanism, particularly in aggregated search, is the first to address in this research. Notably, we provided a balanced architectural approach for information discovery, emphasizing the dataset, data-model, and information diversification equally. We used real-dataset retrieved from the search engines in real-time. We instantiated a non-linear graph data modal consisting of diverse information while preserving the semantics and similarity relationships. Finally, we provided a theoretical background to foster exploration and discovery activities.

Our component-based architecture design includes sub-components. Each sub-component produces a consumable output. There are five main components, referred to as (A) Search Results Aggregation; (B) Multimedia Document Creation; (C) Multimedia Documents Grouping; (D) Graph Instantiation; (E) Semantic Lookup List, components as illustrated in Fig. 2. Each component is concerned with delegated responsibility, and their internal working is separate from each other. A discussion on each component is provided in the following sections.

Figure 2 Discovery Architecture Design.

Component (A) Search Results Aggregation, (B) Multimedia Documents Creation, (C) Multimedia Documents Grouping, (D) Graph Instantiation, (E) Semantic Lookup List.

(A) Search results retrieval & aggregation

Search results retrieved from the search engines are presented in the form of disjoint verticals. Adversely, users’ information needs are becoming complex and multi-modal, requiring the employment of multimedia artifacts for satisfaction. To aggregate scattered disjoint verticals (P1), we introduced a search results aggregation component. The aggregation process of this component is subdivided into three steps.

(i) Vertical retrieval

Definition: We define vertical as a specialized assembly of same-typed search results and retrieval as a process of obtaining them from some external source.

Formalization: Let S be the set of α, β, γ, λ respectively given as S = {α, β, γ, λ}, where α is defined as a set of video snippets V given as α = {{Vψ1, Vψ2, Vψ3, … , Vψn}, {Vϕ1, Vϕ2, Vϕ3, … , Vϕn}}, β is defined as a set of news snippets N given as β = {{Nψ1, Nψ2, Nψ3, … , Nψn}, {Nϕ1, Nϕ2, Nϕ3, … , Nϕn}}, γ is defined as a set of image snippets I given as γ={{I1ψ,I2ψ,I3ψ,...,Inψ},{I1Φ,I2Φ,I3Φ,...,InΦ}}, and λ is defined as a set of web snippets W given as λ = {Wψ1, Wψ2, Wψ3, … , Wψn}, where ψ and ϕ denotes the textual and visual modality associated with a snippet respectively.

Instantiation: We retrieved top hundred search results from each web, news, image, and video verticals. Since exploratory and discovery tasks require increased recall over precision (P4), we chose to retrieve maximum search results from the API provider. With each search result, we preserve the metadata associate with it. The verticals are retrieved from the Google search engine in real-time because Google is highly preferred by web users (Ali & Gul, 2016). The Table 1 shows the vertical retrieval parameters. Figure 3 outlines the possible features of a snippet.

Table 1 Parameters for verticals retrieval.

Vertical	# of results (n)	Source	Modality	Feature(s)	
Web	≤100	Google	Textual	Title, Description, URL	
Video	≤100	Google	Textual + Visual	Title, Description, URL,	
	Thumbnail, Date	
News	≤100	Google	Textual + Visual	Title, Description, URL,	
	Thumbnail, Date	
Image	≤100	Google	Textual + Visual	Title, URL, Thumbnail	

Figure 3 Anatomy of search result snippet.

(A) URI, (B) Title, (C) Description, (D) Date, (E) Thumbnail. Screenshot credit: Google and the Google logo are trademarks of Google LLC.

(ii) Verticals aggregation

Definition: We define verticals aggregation as a single container of all the retrieved disjoint verticals.

Formalization: Let X be the subset of S consisting of the all the elements α, β, γ and λ from S. We consider only textual modality information given as X=∑i=1nSiψ.

Instantiation: Each retrieved snippet has unwanted data (e.g., HTML tags, numbers, special characters, etc.). These impurities do not add meaning to the semantic analysis. We are restraining to perform extra pre-processing steps such as stopwords removal and stemming. It results in the loss of contextual information necessary for semantic analysis. Afterward, we preserve the scattered disjoint verticals textual data inside a single container as a linear list.

(B) Multimedia document creation

Previous studies indicate user interest in exploring multimedia documents encapsulating relevant multimedia objects during information exploration (Rashid & Bhatti, 2017). We define a multimedia document as a semantic container of similar content belonging to multiple modalities. Instead of providing a linear list of snippets, which forces web users to locate scattered relevant multimedia objects from disjoint verticals, we give document-based multimedia exploration (P6). The multimedia document semantically gathers the scattered multimedia objects belonging to various disjoint verticals. This process is again sub-divided into three steps.

(i) Semantic analysis

Definition: We define semantic analysis as a process of obtaining semantic information (relatedness and containment) from transformed multidimensional vector representation of search results textual data (P2).

Formalization: ∀ x ε X let Ex be the set of sentence embedding given as Ex = {e1, e2, e3, … , en}, where each element in Ex is represented by a multidimensional vector e→=(r1,r2,r3,...,r768); rε(R).

Instantiation: Firstly, we transformed each multimedia snippet in the aggregated list into sentence embeddings. This transforms each snippet into a multidimensional vector space for semantic analysis. Since each snippet contains minimal textual description, sentence embedding is deemed a better choice over the Doc2Vec technique.

(ii) Clustering

Definition: We define clustering as a process of grouping the search results, having highly related intra-group coherence and otherwise for the inter-group search results.

Formalization: Ex = c1 ∪ … ci ∪ cn ; ci ∩ cj = 0 (i ≠ j), where Ex denotes original data, ci, cj are clusters of Ex and n is the number of clusters. Let Cd be the set of clusters of Ex given as Cd = {c1, c2, c3, … , cn}, where each cluster contains a set of coherent text t and c = {t1, t2, t3, … , tn}.

Instantiation: We performed the agglomerative clustering on all the semantic search results vectors. Agglomerative clustering is chosen due to flexibility in the clustering process as it allows the clusters to be obtained using cut-off criteria instead of predefined number of clusters. This process groups similar search results in various buckets, called multimedia document.

(iii) Summarization

Definition: We define summarization as a process of extracting the most representative words from the bucket of search results.

Formalization: Let Wd be a set of words sequence w from c ε Cd, generated by text summarizer representing the collection of text given in c as Wd = {w1, w2, w3, … , wn}, let Md be the multimedia document, we formed Md by mapping function Md=∀(Cd)∀(Wd)(f(Cd)=f(Wd)→Cd=Wd).

Instantiation: To enhance sensemaking (P6), instead of merely labeling a multimedia document by assigning predefined categories, we are performing summarization based on the text of the snippets inside the multimedia document. Specifically, we perform extractive text summarization techniques to extract the combination of the most representing text inside the multimedia document for its representation.

(C) Multimedia document grouping

Prior research has shown that web user information exploration behavior is analogous to a foraging animal in the forest (Pirolli & Card, 1999). They look for the patches containing more information scent as compared to the effort performed. In traditional linear list presentation of the search results, a user has extreme difficulty locating the appropriate patches of information and comprehending search results space (Ruotsalo et al., 2015b). This component groups multimedia documents to provide patches of information and enhance search results in space comprehension (P5). This process is sub-divided into three steps.

(i) Semantic analysis

Definition: We define semantic analysis as a process of obtaining semantic information (relatedness and containment) from transformed multidimensional vector representation of multimedia documents.

Formalization: From Wd, we produce the set Y to perform semantic analysis, given as Y=∑i=1nWdi. ∀ y ε Y let Mx be the set of sentence embedding given as Mx = {e1, e2, e3, … , en}, where each element in Mx is represented by a multidimensional vector e→=(r1,r2,r3,...,r768) ; rε(R).

Instantiation: Firstly, we extract summaries of multimedia documents and aggregated them inside a linear list. Then we performed semantic analysis on each multimedia document summary using sentence embeddings. This transformed each multimedia document to a multidimensional vector space for semantic analysis. Similarly, since each multimedia document contains minimal textual description, sentence embedding is deemed a better choice over the Doc2Vec technique.

(ii) Clustering

Definition: We define clustering as a process of grouping the multimedia documents having high intra-group relatedness and otherwise for the inter-group multimedia documents.

Formalization: Let Mx = c1 ∪ … ci ∪ cn ; ci ∩ cj = ø (i ≠ j), where Mx denotes original data, ci, cj are clusters of Mx and n is the number of clusters. Let Cg be the set of clusters of Mx given as Cg = {c1, c2, c3, … , cn}, where each cluster contains a set of coherent text t and c = {t1, t2, t3, … , tn}.

Instantiation: We performed the agglomerative clustering on all the semantic vectors of multimedia document summaries. Similarly, agglomerative clustering is chosen for flexibility in clusters creation process using a cut-off criteria. This process groups similar multimedia documents in various buckets.

(iii) Summarization

Definition: We define summarization as a process of extracting the most representative words from the bucket of multimedia documents.

Formalization: Let Cg be the set of clusters of Mx given as Cg = {c1, c2, c3, … , cn}, where each cluster contains a set of similar text t c = {t1, t2, t3, … , tn} ; tεCg. Text summarizer Wg produces a set of words sequence w from c ε Cg representing the collection of text given in c as Wg = {w1, w2, w3, … , wn}. Similarly, let Mc be the multimedia document cluster, we formed Mc by mapping function Mc=∀(Cg)∀(Wg)(f(Cg)=f(Wg)→Cg=Wg).

Instantiation: We call each generated bucket of multimedia documents from the clustering process a multimedia document group. To enhance sensemaking, instead of merely labeling a multimedia document group by arranging them in taxonomic order, we perform summarization based on the multimedia document summary. The summarization process is performed using extractive text summarization technique. This extracts the most representing text inside the multimedia document group.

(D) Graph instantiation

Present search engines display the search results in a linear list and often ignores the relationship between multimedia content. As a result, users have to navigate the results space and berry-pick the relevant items of interest (Bates, 1989). This results in a non-linear searching pattern of a user in the exploration of information (Bates, 1989). To overcome these challenges, we instantiated a non-linear graph augmenting the users’ non-linear exploratory information-seeking behavior while preserving the relationships (P3). This process is sub-divided into three steps, as well.

(i) Vertices creation

Definition: We define vertex as an atomic data structure encapsulating the complete details of the representing entity.

Formalization: Let a graph G be a set of vertices V and edges E, given as G = (V, E) and vertices V represent all the vertical snippets, multimedia documents and clusters given as V = {S, Md, Mc}.

Instantiation: Firstly, we represented each multimedia document group, multimedia document, and multimedia snippet as a vertex. We associate with each vertex the metadata. It includes a text summary for the multimedia documents and multimedia document groups. Similarly, metadata belonging to the multimedia snippet include their title, description, URI, date, and thumbnail (where available).

(ii) Part-of linking

Definition: We define part-of linking as a process of establishing containment relationship between vertices.

Formalization: The edge (δ) between the S and Md denotes the part-of relationship given as δ : ∀ x ε S, ∃dεMd, f(Md) = S. Similarly, edge (δ) between the Md and Mc denotes the part-of relationship given as δ : ∀mεMd, ∃cεMc, f(Mc) = Md.

Instantiation: Since a multimedia document is a part of some multimedia documents group, similarly, a multimedia snippet is a part of some multimedia document, the edges established between them represents the part-of (or containment) relationship.

(iii) Similarity linking

Definition: We define similarity linking as a process of establishing proximally similarity-based relationship between vertices.

Formalization: Edges (δ) among Mc denotes the similarity relationship based on Cartesian product of Mc given as:

δ:(Mc×Mc=∑i=1n∑j=i+1nJ(Mic,Mjc),if>θ∅,otherwise

Similarly, edges (δ) among Md in each Mc denotes the similarity relationship based on Cartesian product of Md within Mc given as:

δ:(∑k=1nMkc∀MdεMkc:Md×Md=∑i=1n∑j=i+1nJ(Mid,Mjd),ifJ>θ∅,otherwise

where J is the Jaccard similarity defined as J(A,B)=|A∩B||A∪B| and θ is the average similarity score of all the selected vertices pairs in the graph.

Instantiation: Exploratory search also involves navigation of proximally similar multimedia documents in the collection (Savolainen, 2018). It helps a user explore the environment to understand better how to exploit it, selectively seek and implicitly obtain cues about coming steps (Savolainen, 2018). Hence, we provide navigational links to proximally similar multimedia document groups and multimedia documents. These links are established if there is a high proximal similarity between the source and destination vertices. We chose the Jaccard similarity measure because it is computationally less expensive than other similarity techniques (Rashid & Bhatti, 2017).

(E) Semantic lookup list

At present, the aggregation of the verticals on the major search engines is provided as partially-blended. The relationship between the multimedia snippets in those retrieved disjoint verticals is ignored (Rashid & Bhatti, 2017). On the other hand, information lookup is an eminent component of information exploration and discovery, and linear lookup lists have proven to be effective in information lookup (Tablan et al., 2015). To overcome this challenge of disjoint and relation-less aggregation of the verticals and provide ease in lookup searches, we introduce a semantic lookup list component that fully-blends the disjoint verticals using semantics of the multimedia snippets (P1). This component consists of two steps.

(i) Similarity calculation

Definition: We define similarity calculation as a process of extracting numeric similarity score between pairs of text using a textual similarity measure.

Formalization: Let the Ex be the same set of sentence described previously, we also transformed user query Q as a sentence embedding Qx represented by Qx→={r1,r2,r3,...,r768}; rε(R). We calculated similarity as Ls={∀e→εEx|0≤SIM(Qx→,e→)≤1}, using a cosine similarity measure defined as SIM(Qx→,e→)=Qx→.e→‖Qx→‖×‖e→‖.

Instantiation: In this part, we transform the user query itself into the sentence embeddings. This transformation eliminates the data representation gap. We perform a similarity calculation operation on the pair-wise (query and each snippet embedding) obtained semantics using a cosine similarity measure. We use cosine similarity because our query and search results are in vector representation.

(ii) Re-ranking

Definition: We define re-ranking as a process of arranging search results in descending order of query and search results embedding pairwise intra-similarity scores.

Formalization: Using similarity scores Ls, we define Lr the ranked linear search results list, sorted in descending order of similarity, given as Lr = {l1, … ,l|X| | f(li) ≥ f(lj), i < j ≤ |X|, li,lj ∈ X }, where f: X → Ls.

Instantiation: In lookup searches, the ordering of information is mandatory. The most relevant information must be present on the most top. The search engines return disjoint ranked verticals. To calculate the ranking order for snippets belonging to aggregated disjoint verticals, we re-rank each multimedia snippet in their descending order of similarity, allowing the most relevant snippet to appear first on the linear list.

Architectural implementation

We implemented our architecture in Python 3 programming language using publicly available libraries. Search results are retrieved using freely available APIs to fetch the verticals from a search engine in real-time. We used Google search engine to retrieve the search results belonging to the web, news, image, and video verticals. We preserved the metadata associated with each snippet, such as the URL, title, date, length, description, and thumbnail, where available. For text summarization, we used LexRank (https://gist.github.com/rodricios/fee45381356c8fb36004/) extractive text summarization algorithm. Semantic analysis is done using SBERT’s (https://pypi.org/project/sentence-transformers/) sentence embedding on pre-optimized bert − base − nli − mean − tokens (https://github.com/UKPLab/sentence-transformers/) pre-trained modal and agglomerative clustering using the ward’s linkage method from sklearn (https://scikit-learn.org/) python library to obtain the clusters. We used Networkx (https://networkx.github.io/) python library to instantiate an undirected network to build the graph. Each node represented either a web snippet, multimedia document, or a multimedia document cluster. The snippet nodes attribute includes their metadata. The multimedia document and multimedia document cluster nodes attribute include their summarized text. Figure 4 shows the visualization of the instantiated graph generated from Cytoscape (https://cytoscape.org/).

Figure 4 Visualization of the instantiated graph.

The orange and pink color represents cluster and multimedia document respectively. The rest denotes snippets belonging to disjoint verticals.

Results

There is still no standard empirical evaluation measures for evaluating the aggregated search approach effectiveness (Li et al., 2017). These approaches are mostly considered in terms of the achieved precision & recall (Rashid & Bhatti, 2017) and judgment reports from the human experts (Ruotsalo et al., 2018). Calculating precision and recall in our case is a non-trivial task. It is mainly due to the nature of the data. Therefore, we used a real dataset with no prior labeling by human experts. Our empirical evaluation measures mostly depend on metrics requiring no initial labeling of data. We used internal clustering stability measures to evaluate the internal cluster model stability (Wani & Riyaz, 2016), and clustering accuracy based on the judgment of the human experts (Ruotsalo et al., 2018). We obtained accuracy and stability scores by dispatching pre-defined queries on Google’s real dataset.

We collected queries from the recently published ORCAS (Craswell et al., 2020) dataset consisting of 10 million distinct records. Selecting all queries in the dataset for evaluation purposes was not practical. Hence, we performed bi-gram and tri-gram query analysis on the ORCAS dataset. Afterward, we selected 25 queries from the top 100 most repeating bi-gram and tri-gram combinations. The average query length for this evaluation was set to 2.5 words. The chosen length was due to a recent study in (Degbelo & Teka, 2019) indicating average user query length between 2.44 and 2.67 words, which confirms that users’ information needs are becoming exploratory. Since in exploratory search, user needs are ambiguous, and the primary objective is to gain an overview of the information. Users type short queries instead of well-articulated longer queries as in the lookup search scenarios (Athukorala et al., 2016). We selected queries covering broad aspects. Therefore, an average query length of 2.5 words was considered based on the average of 2.44 and 2.67 words.

Internal clustering parameterizing

We used agglomerative clustering for the creation of multimedia documents and multimedia documents groups. We specified cut-off threshold criteria for the cluster creation process θ to form the desired number of clusters. We chose θ empirically by determining the best possible average mean value of internal cluster stability measures. We used a well-known Rousseeuw (1987) Silhouette Coefficient (SC), Caliński & Harabasz (1974) Index (CHI) and Davies & Bouldin (1979) Index (DBI) to calculate internal cluster stability. We calculated the mean average value of θ1 by performing five experiments and taking their mean value to create multimedia documents. Based on the obtained θ1 threshold, we again repeated the same procedure for multimedia documents clustering to obtain θ2. This process of obtaining θ1 and θ2 is displayed in Table 2. Finally, we parameterized the clustering model for multimedia documents and documents clusters based on empirically obtained values, as displayed in Table 4.

Table 2 Empirical clustering results for (1) Multimedia Documents (2) Multimedia document groups.

Experiment	Iteration	Optimal θ1	# of Clusters1	SC1	CHI1	DBI1	Optimal θ2	# of Clusters2	SC2	CHI2	DBI2	
	1	15	174	0.11	3.40	1.15	17	79	0.08	4.97	1.79	
	2	9	200	0.15	5.88	0.88	19	5	0.05	3.15	2.00	
1	3	12	240	0.12	3.82	0.83	14	90	0.05	2.28	1.06	
	4	14	148	0.11	3.59	0.98	17	54	0.05	2.66	1.47	
	5	10	248	0.15	5.60	0.75	13	54	0.07	3.32	1.08	
	Mean	12	202	0.13	4.46	0.92	16	56.4	0.06	3.28	1.48	
	1	14	209	0.10	3.39	0.97	17	76	0.04	2.83	1.81	
	2	15	157	0.09	3.56	1.27	16	81	0.04	2.61	1.42	
2	3	15	160	0.11	3.36	1.28	15	113	0.05	2.51	1.21	
	4	16	123	0.08	3.62	1.50	15	93	0.05	2.54	1.24	
	5	13	255	0.15	4.08	0.74	16	97	0.06	2.73	1.28	
	Mean	14.6	180.8	0.11	3.60	1.15	15.8	92	0.04	2.65	1.39	
	1	13	187	0.08	3.49	1.08	15	61	0.05	2.93	1.58	
	2	15	180	0.10	3.36	1.11	16	106	0.04	2.57	1.33	
3	3	13	191	0.10	4.01	1.01	15	69	0.05	2.84	1.27	
	4	16	102	0.09	4.51	1.59	15	77	0.04	2.69	1.24	
	5	15	178	0.12	3.48	1.12	16	103	0.05	2.54	1.29	
	Mean	14.4	167.6	0.10	3.77	1.18	15.4	83.2	0.05	2.71	1.34	
	1	14	190	0.12	3.94	1.01	16	82	0.06	2.63	1.39	
	2	14	160	0.10	3.80	1.20	16	55	0.06	2.84	1.56	
4	3	14	205	0.09	3.06	1.06	16	85	0.04	2.64	1.43	
	4	13	249	0.12	3.22	0.81	15	114	0.04	2.30	1.16	
	5	13	186	0.17	4.47	0.88	15	78	0.05	2.58	1.18	
	Mean	13.6	198	0.12	3.70	0.99	15.6	82.8	0.05	2.60	1.34	
	1	14	166	0.10	4.04	1.16	15	78	0.04	2.76	1.27	
	2	13	192	0.09	3.85	1.00	16	80	0.05	3.01	1.62	
5	3	13	182	0.10	3.58	1.09	15	58	0.05	2.78	1.40	
	4	12	204	0.10	4.06	0.96	14	60	0.06	3.01	1.13	
	5	12	192	0.14	5.46	0.92	14	64	0.05	2.74	1.14	
	Mean	12.8	187.2	0.11	4.20	1.03	14.8	62	0.05	2.86	1.31	
Mean Average		13.48	187.12	0.11	3.95	1.06	15.52	75.28	0.05	2.82	1.37	

Table 3 Clustering precision.

Clusters precision for (1) Multimedia documents, (2) Multimedia document groups, Relevancy scores by (a) Novice judge (b) Expert judge.

Experiment	Iteration	Precision %age (1a)	Precision %age (1b)	Precision %age (2a)	Precision %age (2b)	
	1	100.00	100.00	100.00	92.00	
	2	100.00	100.00	100.00	100.00	
1	3	100.00	96.70	100.00	100.00	
	4	96.60	88.00	100.00	100.00	
	5	100.00	100.00	100.00	100.00	
	Mean	99.32	96.94	100.00	98.40	
	1	100.00	100.00	100.00	100.00	
	2	100.00	100.00	100.00	100.00	
2	3	100.00	100.00	100.00	100.00	
	4	100.00	100.00	100.00	100.00	
	5	100.00	100.00	100.00	100.00	
	Mean	100.00	100.00	100.00	100.00	
	1	100.00	100.00	100.00	100.00	
	2	100.00	100.00	100.00	100.00	
3	3	100.00	100.00	100.00	100.00	
	4	100.00	100.00	100.00	100.00	
	5	96.60	100.00	100.00	98.20	
	Mean	99.32	100.00	100.00	99.64	
	1	98.10	100.00	98.20	100.00	
	2	100.00	100.00	96.90	100.00	
4	3	96.90	96.90	100.00	100.00	
	4	100.00	100.00	100.00	100.00	
	5	100.00	100.00	100.00	100.00	
	Mean	99.00	99.38	99.02	100.00	
	1	100.00	100.00	100.00	100.00	
	2	100.00	100.00	100.00	100.00	
5	3	100.00	96.60	100.00	100.00	
	4	100.00	100.00	100.00	100.00	
	5	100.00	100.00	100.00	100.00	
	Mean	100.00	99.32	100.00	100.00	
Mean Average	99.53	99.13	99.80	99.61	
Average	99.33	99.71	
Cohen’s Kappa	0.474	0.398	

Table 4 Clustering model parameters for (1) Multimedia documents (2) Multimedia document groups.

Parameters	Description	Value1	Value2	
n_clusters	# of clusters to find	None	None	
affinity	Metric to compute linkage	Euclidean	Euclidean	
distance_threshold	The linkage distance threshold for merging clusters	13.48	15.52	
linkage	Distance method between set of observations	Ward	Ward	

Clustering precision

Precision is referred to as a fraction of relevant retrieved out of total relevant results (Rashid & Bhatti, 2017). In clustering, precision is a fraction of relevant results out of total results inside a cluster. Precision is mostly calculated by cross-matching obtained cluster results with correct labeled data. In a real dataset, the labeling of data is unavailable. We logged the search results retrieved from the pre-defined queries during the empirical internal clustering model parameterization process to overcome this challenge. These logged search results were then presented to two human experts to label relevant and irrelevant search results inside each cluster. The existing literature mostly use two human experts for verification of the judgments between the experts (Achsas & Nfaoui, 2018; Koh et al., 2006; Taramigkou, Apostolou & Mentzas, 2017). To perform the rigorous expert-based evaluation, we hired two human experts from opposite backgrounds, we then statistically measured their amount of agreement. To the best of our knowledge, this human-expert background diversity and their inter-relevance agreeability has not been considered previously. The first human expert is a graduate in education and had no prior knowledge about computing-related technical aspects. The second human expert is a graduate in computer science and had substantial knowledge about computing technical aspects, including the concept of clustering. This diversity in the background helps in obtaining unbiased validation of our clustering approach.

Table 3 show the results obtained from the human experts. We run a total of 25 experiments, divided into 5 iterations. From each iteration, we obtained the mean results. Afterward, we took the mean average of 5 iterations. This process was repeated for both; the multimedia documents and multimedia document groups. The results show no significant change in the relevancy judgment scores from both the novice judge (99.53%) and the expert judge (99.13%) for multimedia documents. Similar results were achieved for multimedia document groups from the novice judge (99.80%) and expert judge (99.61%). There is a moderate amount of agreement (κ = 0.474) between the novice and expert judges for multimedia documents. Similarly, there is a fair amount of agreement (κ = 0.398) between the novice and expert judges for multimedia document groups. Since there was a very low average standard deviation (SD = 0.15) in the obtained relevancy scores and there also existed a fair to moderate amount of agreement, the relevancy report from the two human experts is deemed comprehensive and complete.

Comparison & discussion

Our approach outperforms in terms of accuracy (99%) in comparison to the approach provided by Achsas & Nfaoui (2018) (89%). It mainly can be due to variations in the data used for model training, choice of deep learning model, and parameterization process. We have performed rigorous and statistically significant empirical evaluation using average scores from human experts having diverse backgrounds and internal clustering stability measures. It presents as a baseline, and a promising start for future search results aggregation approaches. This signifies that the previous researches in this domain are excessively concerned with innovations in the existing techniques, which we believe, is unnecessary while the present search engines are still suffering from the major issues discussed in this paper. We also made an effort to emphasize the fact that the existing techniques are sufficiently optimized to solve real issues and there exists a need for attention from the researchers towards more emerging trends.

Each research utilizes different techniques and mechanisms to provide information exploration and discovery. We have extracted the major parameters and their possible values for an in-depth comparison of our approach with existing state-of-the-art. To ease comprehension of these parameters, we have further categorized parameters according to their purpose, as displayed in the Table 5. The provided functionalities are marked with the “+” symbol, whereas the missing functionalities are left blank.

Table 5 Comparison of the proposed approach with state-of-the-art.

Category	Parameter	Values	State-of-the-Art	
			(Koh et al., 2006)	(Ruotsalo et al., 2015a)	(Tablan et al., 2015)	(di Sciascio, Sabol & Veas, 2016)	(Krishnamurthy et al., 2016)	(Khalili et al., 2017)	(Lisena et al., 2017)	(Rashid & Bhatti, 2017)	(Taramigkou, Apostolou & Mentzas, 2017)	(Zhang et al., 2018)	(Kanjanakuha, Janecek & Techawut, 2019)	Proposed Approach	
	Search Type	Full text	+	+	+	+	+	+		+	+	+		+	
		Fielded			+			+	+						
Searching		Semantic			+			+	+				+	+	
		Federated	+		+		+			+	+			+	
	Search Results Granularity	Snippets	+				+	+	+		+		+	+	
		Document		+	+	+						+		+	
		Document Clusters												+	
	Search Activity	Lookup							+			+		+	
		Exploratory	+	+	+	+	+	+		+	+	+	+	+	
		Discovery	+	+	+		+	+	+		+		+	+	
	Information Source	Web	+				+				+	+		+	
		Repository	+	+	+	+	+	+	+	+	+		+		
		Real	+				+				+			+	
Data	Data Model	Linear		+	+	+	+				+			+	
		Non-linear	+		+			+	+	+		+	+	+	
	Data Relation	Part-of			+			+	+	+			+	+	
		Similarity	+	+	+	+	+	+	+	+	+	+		+	
		Semantic			+		+	+						+	
	Media Source	Textual	+	+	+	+	+	+		+	+	+	+	+	
Information Retrieval		Multimedia	+						+	+	+			+	
	Retrieval Modal	Monomodal		+	+	+	+	+				+	+		
		Cross-Modal	+						+	+	+			+	

Table 5 emphasizes the three significant aspects of discovery techniques. The first aspect is searching for search results, including search type, search results granularity, and searching activity. The second aspect concerns data management, including information sources, instantiation of data-modal, and assembling mechanism. Finally, the third aspect is concerned with technical information retrieval aspects of the discovery and exploratory approaches, including media sources and information retrieval modal.

The most crucial factor in information discovery is flexibility in representing the information to avoid information overload. Most of the existing research solely relies on filtering capabilities but lacks in providing appropriate granularity control of the search results (di Sciascio, Sabol & Veas, 2016). Our approach provides three-level granularity; snippets, multimedia documents, and multimedia documents clusters. The data-modals employed by the existing researches are mostly centered around a specific domain and specific data. They mainly include the scientific domain having millions of literature as a dataset. Approaches providing real datasets were also primarily concerned with integrating a few verticals such as web and image (Koh et al., 2006). To enable our approach to be generic and applicable to all the domains and datasets, we presently use only real datasets to observe our approach’s integrity even in the most variate and uncertain data coming from the search engines in real-time.

Information exploration and discovery is a long, non-trivial, and non-linear journey. To foster non-linear navigation of the search results, existing literature mostly instantiated a graph data-modal using either existing domain knowledge, such as ontologies (Khalili et al., 2017; Lisena et al., 2017; Kanjanakuha, Janecek & Techawut, 2019), or using some generic similarity measures (Rashid & Bhatti, 2017). Our approach uses domain-independent semantics and similarity measures to construct a non-linear graph to provide non-linear means of search results exploration and discovery.

Information management is also an essential factor in enabling information discovery and a compelling exploration of the search results. Numerous information management approaches organize and present the users’ search results, increasing their cognitive abilities. These approaches include linear and non-linear browsing of information and summarization. However, previously, these were implemented as disjoint components, combined on a single interface (Fung & Thanadechteemapat, 2010). Our approach unifies all of the specific techniques and encapsulates it in a single component. With our generic information discovery architecture based on a strong theoretical background and promising empirical evaluation results, we hope to provide a new baseline for future researches on relational aggregated search and search engines alike.

Conclusion & future work

In this research, we proposed a generic discovery architecture using multimedia search engine results. A brief discussion on information exploration and theoretical discovery background was provided, and an architectural solution was formalized and instantiated. Before this work, the exploration and discovery of information on the web search engine were leveraged using traditional heuristics. We identified potential gaps and issues in the current general web search engine approach. To overcome these issues, we presented a new baseline using search results aggregation. Our approach was employed using state-of-the-art sentence embeddings. We bridged the gap between the abundant multimedia contents by encapsulating semantically multimedia artifacts in multimedia documents and summarizing them. Moreover, we eased the navigation problem in the search results space by grouping multimedia documents in semantically similar patches.

The proposed discovery architecture emphasizes all the aspects of the discovery, including information exploration and lookup. We supported information exploration by providing the nonlinear proximal navigation and exploration support through the instantiation of a complex graph and lookup searches through a semantically fully-blended ordered linear search results list. Finally, a comprehensive empirical evaluation was presented. The empirical evaluation out-performed previous aggregation approaches at all granularity levels of aggregation provided in this research. To the best of our knowledge, our approach is the first to be assessed comprehensively from the system and the user perspective on the dataset and the queries obtained from the user and the search engine, respectively, in real-time.

We believe that the user experience and the user interface both play an important role to influence information discovery. This aspect requires a detailed and comprehensive discussion, which was out of the scope of this research. In the future, we look forward to providing a more ablative evaluation from the usability perspective of architecture involving an even broader audience with extremely varied backgrounds and experiences with more focus on human aspects, including user interfaces. We have intentions to provide the adaptable clustering of multimedia documents by considering the users’ diverse information need and information-seeking behavior. We are interested in investigating multimodal verticals aggregation and exploiting various nonlinear data models consisting of multiple modalities in the enhanced discovery of aggregated multimedia based document search results in real-time scenarios.

Supplemental Information

Supplemental Information 1 Code logic and Query cached data.

We used each query cached file as the single experimental run. We calculated the mean from 5 iterations and again calculated the mean average of 5 iterations in the end.

Click here for additional data file.

Additional Information and Declarations

Competing Interests

Author Contributions

Data Availability

The authors declare that they have no competing interests.

Abdur Rehman Khan conceived and designed the experiments, performed the experiments, analyzed the data, performed the computation work, prepared figures and/or tables, authored or reviewed drafts of the paper, and approved the final draft.

Umer Rashid conceived and designed the experiments, prepared figures and/or tables, authored or reviewed drafts of the paper, and approved the final draft.

Khalid Saleem conceived and designed the experiments, authored or reviewed drafts of the paper, and approved the final draft.

Adeel Ahmed conceived and designed the experiments, authored or reviewed drafts of the paper, and approved the final draft.

The following information was supplied regarding data availability:

Code, logic, and caching query files used for experimental purposes are available in the Supplemental Files.

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
