# Peer review of "An architecture for non-linear discovery of aggregated multimedia document web search results"

_PeerJ Computer Science, doi:10.7717/peerj-cs.449_

## Round 0.1 · original submission · Minor Revisions

The reviewers mention that the paper is well-written and there is a well-designed evaluation of their approach. I recommend that the paper be accepted with minor revisions to address the formatting issues in tables and formulae. The architecture section should be more concise.

No further development/experimentation is required for this manuscript, but the authors should refer to the comments of reviewer 2 as potential directions for future work.

The work makes a contribution to search engine research and is worthy of publication with these changes.

·

Basic reporting

The research study described in the article is very well structured and focuses on evaluating the benefits of aggregating "multimedia" information by "verticals". The formal definition of "vertical" (page 7) includes video snippets, news snippets, image snippets, and Web snippets. In this context, I believe that the usage of the term "multimedia" is somewhat misleading. The actual study processes the metadata retrieved from a Web document, which is represented by a Web page, a video, news etc. The metadata of each Web document includes the title, the description, the date, and the URL (a thumbnail if present.) Although this information is retrieved from documents that include multimedia content (e.g. videos, images) the authors use mainly the text-based information related to each Web resource. For this reason, I would suggest to reconsider the usage of the term "multimedia" and maybe substitute it with an expression that represents more the metadata of various Web resources.

Experimental design

The design of the experiment is outlined in great detail. Considering the challenge of the evaluation, the authors always support their choices either providing specific information (e.g. average query length that was used, page 12) or referencing other research studies.
The architecture of the experiment is based on well-known clustering techniques as well as standard software libraries.

The authors chose 2 users from different backgrounds (i.e. education and computer science) to conduct the evaluation of their experiment. I believe that this evaluation should be conducted by more than 2 human experts.

On page 5 the authors mention about “disjoint components” as one of the problems on the Google SERP page. I think that this aspect should be given more attention, especially during the design of the evaluation. Both the user experience and the user interface play an important role to influence the discovery techniques of the users related to “relevance”. This aspect is partially mentioned in the conclusions.

Validity of the findings

Although the techniques used in the study are not new, their application along with the evaluation process of the study are very interesting.
The value of accuracy (i.e. 99%) in the findings outperforms by far other approaches that are mentioned.

This experiment uses snippets, multimedia documents, and multimedia documents clusters. As shown in table 7, it seems to outperform other related studies when applied to discovery techniques. These findings provide a new, solid baseline for future experiments focusing on information exploration and discovery by search engines users.

Reviewer 2 ·

Basic reporting

- The paper is clearly written. However, the paper is a little bit long and wordy.

- The authors provide sufficient literature review and background.

- The structure of paper is well organized, figures look fine. However, texts in tables are not consistent. Several tables have large font size (larger than the font size of content).

- The results support the hypothesis proposed by authors.

Experimental design

- The authors propose a method for aggregating search result by using semantic analysis. In particular, the authors enhance the search engine shortcomings in information exploration and discovery activities by augmenting non-linear information seeking patterns. From the initial search results, the proposed method represents information in the semantically high-dimension space followed by performing clustering, semantic analysis, summarization, then linking information by a graph.

- The authors conduct their experiments on ORCAS dataset. The experiments are measured based on the clustering stability measures and clustering accuracy based on human experts and novices.

- The experimental results have shown the improvement the proposed method compared to prior methods.

Validity of the findings

- The proposed method sound reasonable and interesting. The authors address the realistic, necessary problem in current search engines. The experimental results have shown the performance of the proposed method. Also, the experimental results partly support the hypothesis and address the problem proposed by the authors.

- However, the paper lacks novelty. The method is incremental since the author incorporate prior techniques to be the proposed method.

- The authors claim that they will provide theoretical background to foster exploration and discovery activities. However, these look like explanations rather than the theoretical proofs. The authors should conduct more ablative experiments to prove their claim in the proposed method section.

Additional comments

- The paper should present in a concise form. In the architecture design, the authors should rewrite this section; It is hard to follow this section.

- The mathematical symbol should be edited. For example, the vector or matrix should be bold (use \mathbf).

- The authors should modify tables, the font size should be consistent with the font size of content.

---

## Round 0.2 · accepted · Accept

The authors have addressed the editor and reviewer comments and the paper is improved and ready for publication.